# Parkinson’s Disease: Exploring Different Animal Model Systems

**DOI:** 10.3390/ijms24109088

**Published:** 2023-05-22

**Authors:** Engila Khan, Ikramul Hasan, M. Emdadul Haque

**Affiliations:** 1Department of Biochemistry and Molecular Biology, College of Medicine and Health Sciences, United Arab Emirates University, Al Ain P.O. Box 15551, United Arab Emirates; 2Department of Pharmaceutical Technology, Faculty of Pharmacy, University of Dhaka, Dhaka 1000, Bangladesh; 3Zayed Center for Health Sciences, United Arab Emirates University, Al Ain P.O. Box 15551, United Arab Emirates

**Keywords:** α-synuclein, animal model, dopaminergic neurons, Lewy body, motor impairment, neurodegeneration, oxidative stress, Parkinson’s disease

## Abstract

Disease modeling in non-human subjects is an essential part of any clinical research. To gain proper understanding of the etiology and pathophysiology of any disease, experimental models are required to replicate the disease process. Due to the huge diversity in pathophysiology and prognosis in different diseases, animal modeling is customized and specific accordingly. As in other neurodegenerative diseases, Parkinson’s disease is a progressive disorder coupled with varying forms of physical and mental disabilities. The pathological hallmarks of Parkinson’s disease are associated with the accumulation of misfolded protein called α-synuclein as Lewy body, and degeneration of dopaminergic neurons in the substantia nigra pars compacta (SNc) area affecting the patient’s motor activity. Extensive research has already been conducted regarding animal modeling of Parkinson’s diseases. These include animal systems with induction of Parkinson’s, either pharmacologically or via genetic manipulation. In this review, we will be summarizing and discussing some of the commonly employed Parkinson’s disease animal model systems and their applications and limitations.

## 1. Introduction

More than two centuries ago, English physician James Parkinson first reported a clinical syndrome having involuntary tremulous motion along with decreased muscular power in an article titled ‘An Essay on the Shaking Palsy’. The syndrome later came to be named after James Parkinson as Parkinson’s disease (PD) [1,2]. PD is a chronic neurodegenerative disorder usually characterized by substantial reduction of dopaminergic neurons in the SNc region and presence of Lewy bodies (which are intracytoplasmic inclusions of proteins—α-synuclein and ubiquitin—and a major histopathological hallmark of the disease). The ultimate expression of this neurodegeneration is abnormal motor symptoms. Bradykinesia, postural instability, muscle tone rigidity, resting tremor, and gait abnormalities are some of the unusual motor symptoms that arise as a result of this neurodegeneration (together these symptoms are referred as parkinsonism/parkinsonian syndrome) [3]. In addition to these, some non-motor symptoms that also manifest include disturbances in sleep, dementia, sensory and autonomic dysfunction, and abnormalities such as constipation, pain, depression, and inability to smell.

Various research evidence supports and suggests the interaction of different factors such as aging, genetics, or even the environment in the prognosis of PD, describing the key pointers leading up to it as multifactorial. Some underlying disease processes include dysregulation of cellular proteostasis, mitochondrial dysfunction, neuronal inflammation, oxidative stress, and lysosomal and autophagy failures. PD may be classified into sporadic or familial subtypes. Dopaminergic, which is loss observed with familial PD, includes mutations in genes encoding for SNCA (α-synuclein), Parkin/PARK2, ubiquitin carboxy terminal hydrolase-1 (UCHL1), PINK1, DJ-1/PARK7, and LRRK2 [4,5]. Normally, α-synuclein is abundantly present in the presynaptic terminals in different forms (oligomeric/monomeric/aggregated; it is an intrinsically disordered protein type) and functions to regulate synaptic vesicular trafficking and neurotransmitter release [6,7,8,9,10,11,12]. In the case of PD, these proteins misfold and aggregate, resulting in the formation of Lewy bodies/Lewy neurites that start spreading in the brain similar to prions. Different pre-clinical studies have suggested that α-synuclein-induced inclusion toxicity is the main player of dopaminergic neuronal death. Current treatment regimens include the use of levodopa or other dopaminergic agonists that relieve patients from symptomatic motor issues via restoration of the neurotransmission process, but, in most cases, this intervention comes up with unwanted severe side-effects and complications. Unfortunately, there are no treatments as such that have been shown to halt the neurodegenerative process in patients [3].

Due to the complexity in the etiology and multifactorial nature of PD, studying the disease mechanism and finding an ideal disease model is of utmost importance. The past few decades have seen significant discoveries and breakthroughs in disease modeling which have been possible via the use of various animal and cellular models. Of course, considerable advancements have been achieved in modeling PD, and, still, work is under way to reach a potentially ideal model which could help us attain significant therapeutic successes. The aim of this review is to highlight the existing experimental model systems for PD, their pros and cons, future perspectives, and outstanding questions that remain to be answered.

## 2. Parkinson’s Disease Model Systems

The rationale behind the use of an experimental model system to emulate the PD phenotype is to explore and discover potential therapy and treatment, and gain further understanding about the disease progression. Studying such model systems presents a platform to identify possible new therapeutic targets for disease intervention. Over the past few years, researchers have achieved more clarity and understanding about the genetics, pathology, and disease progression and heterogeneity of PD owing to the use and application of different experimental models.

Presently, the existing PD models capture the disease’s pathology partially. A factor contributing to this could be the fact that ‘model systems’ should, ideally, be able to develop the disease pathology in a relatively shorter span of time for us to study it well, unlike the duration of actual PD development in humans [13]. Bearing in mind the variations and heterogeneity in the cause and origin of PD, efforts have been made to model the disease pathology, aiming at recapitulating α-synucleinopathy observed in PD, genetic types of PD, and dysfunction in the midbrain dopaminergic neuronal signaling model systems (via toxin or other pharmacological interventions). These model systems usually represent certain attributes of PD, such as observed changes in behavior, electrical activity, and changes at the cellular or molecular levels [14].

PD experimental model types may be classified into animal- and cell-culture-based model systems (Figure 1). The disease model systems may use environmental or synthetic neurotoxins or a genetics-based approach to study the disease pathology. Each group comes with its own set of merits and demerits, and, therefore, learning about the existing variations and how they are used to study PD may enable researchers to decide correctly when selecting an appropriate model system for their specific experiments. The traditional toxin-based animal models of PD which were developed via destroying the dopaminergic neurons assisted in treatment development for PD symptoms and investigating the adverse effects that might be related to therapies involving dopamine replacement. However, these have not been able to modify, reduce, or reverse the disease’s course. Other model systems looking at α-synuclein-induced dysfunction and death (rather than just direct neuronal death) are closer to modulating the disease’s chronic degenerative procession [1].

The key features that cellular models have been observed to replicate include dopaminergic neuronal degeneration and the presence of α-synuclein protein aggregates. While cellular models may be more advantageous than the in vivo animal models in terms of cost-effectiveness, time, and ease, appropriate model selection depends on the particular aspect of PD [13].

Taken together, exploring different experimental models and ways of PD induction forms an essential part when deciding what type of outcomes are expected and which one modulates the pathology closest and is most relevant to the investigation at hand [15].

## 3. Animal Model

For any Parkinson’s disease model to hold value, certain general features or characteristics of the model must be laid down. According to these, one can judge or determine how close a disease model is to the actual situation and take better decisions while analyzing the results. The following are some desirable pointers [15,16] that an ideal animal model should have:The presence of a complement of DA neurons during the birth stage, with specific and gradual depletion of DA neurons as the organism progresses to adulthood. The loss in neurons should be more than 50% of the total amount and be easily noticeable via biochemistry- and neuropathology-related techniques [16];The model animal must be able to exhibit motor deficits observed in the disease or the expected behavioral phenotype, including slowness in movement, resting tremor, and rigidity [16];The presence of Lewy body and its development as an indicator of the manifestation of α-synuclein pathology [16];The model must also be sure to replicate the disease progression over a period of a few months allowing for a faster and less expensive screening of potential therapeutic candidates [16].

## 4. Common Laboratory Animals Used to Model PD

### 4.1. Rodents

Rodents are among the most popular animal models used across research groups, given the ease of handling and care required. They do not need a special, hard-to-achieve set-up for breeding and management. They are small-sized animals whose anatomy is relatively similar to humans to a certain extent. A classical animal species is used for a PD model system. Specifically, rats or mice are widely used to model PD due to the correlation between motor dysfunction/deficit and dopaminergic neuronal degeneration in the SNc. In these animals, PD can be induced pharmacologically, or via specific genetic manipulation, and these are broadly known as transgenic rodents.

Pharmacological or toxin-based induction usually lacks molecular similarities with parkinsonism in humans but these are useful in developing motor, non-motor, and behavioral aspects of PD [17,18]. Some of these aspects include bradykinesia (measured by pole test) [19], locomotor activity (measured by open field test) [20], akinesia (measured by stepping test) [21], strength, balance, and coordination (measured by rotarod test) [22], observing and monitoring daily activity of the animal (such as drinking, sleeping, and eating), and the presence of any compulsive behavior or lack of motivation [1,23,24].

On the other hand, familial PD may be more accurately simulated using genetic models. Though genetic interventions can induce different molecular dysfunctions (such as dysfunctional mitochondria [25,26,27], altered mitophagy [28,29], ubiquitin proteasome dysfunction [30], and altered ROS production [31]) that are related to PD, they lack important pathological manifestation of PD-like presence of Lewy bodies, loss of DA neurons, etc. [32].

### 4.2. Non-Human Primates (NHPs)

NHPs bear a close relation with human beings in terms of genetic makeup and physiology [33]. It is for this reason that they have come to fulfil an essential part in helping gain better understanding and insights into the underlying disease mechanisms. However, due to ethical issues, expenses, and the amount of effort and labor required for management, their use is very limited, but studies using such animals for investigating PD can be conducted if required for pre-clinical therapeutic evaluation [34]. Commonly used NHPs to model PD include macaques [35], marmosets [36], squirrel monkeys [37], baboons [38], and African green monkeys [39].

Similar to other animal models, PD can be induced in NHPs by neurotoxin administration or by genetic intervention [34]. Commonly used neurotoxins in NHPs are MPTP [40] and 6-OHDA [41]. Genetic interventions for inducing PD in NHPs include injection of viral vectors encoding for overexpression of α-synuclein and LRRK2 (autosomal-dominant genes) and knockout or knockdown models for PINK1, Parkin, and DJ-1 (autosomal-recessive genes) [42,43,44,45,46]. Different motor and neuropathological hallmarks of Parkinson’s disease were reported in monkeys and macaques after adeno-associated virus (AAV) vector-based genetic insertion of human α-synuclein gene having mutation at A53T [43,47,48].

NHP animal models display disease symptoms such as those observed in humans (such as chorea/dystonia) [1]. Apart from those, they even have a similar sleeping pattern/cycle as that of humans. With respect to these aspects, NHPs are much better and superior animal models when compared to rodents. Neuro imaging studies and analyses have shown that the NHP animal models are highly trustworthy and provide essential knowledge or information [34]. Lewy bodies, which are among the major histopathological hallmarks of PD, can only be observed in NHPs as compared to other models. While the contribution of NHPs is of great significance in PD animal model systems, they are restricted in terms of the high level of skills, expertise, support, and time that is required [1,49].

### 4.3. Non-Mammalian Species (NMSs)

This group consists of small organisms such as *C. (Caenorhabditis) elegans*, zebrafish, *Drosophila melanogaster*, etc. Properties such as low maintenance cost and short lifespan render these organisms ideal for research mostly involving genetic/gene manipulations [50]. Among their most important features are the exhibition of clearly defined neuropathology and observable behavior. NMSs are especially useful in whole-genome sequencing experiments and large-scale drug screening.

#### 4.3.1. *Caenorhabditis elegans*

*C. elegans* are host of a network of 302 neurons and 8 dopaminergic neurons [51]. Expression of homologues of various human genes, including LRRK2 (lrk-1), PINK1 (pink1), PARKIN (pdr-1), and DJ- 1 (dnaj-1.1, dnaj-1.2) (but not the α-synuclein), that are implicated in familial PD is observed in this organism [52]. This nematode model was used in the late 1900s (1970s) to investigate the underlying genetic base for neuromuscular activity by Sydney Brenner [53]. The fact that it has only eight ‘anatomically defined’ dopaminergic neurons allows for great accuracy in quantifying neurodegeneration that may have been affected by external/internal stress factors [52].

#### 4.3.2. *Drosophila melanogaster*

*Drosophila* has a well-defined nervous system, with the adult brain consisting of a dopamine-producing neuronal cluster comprising around 200 DA neurons [53]. This renders it a good model system in which we can recapitulate and study PD-associated neurodegeneration. PD symptoms, such as those of dopaminergic neuronal loss, formation of inclusion bodies, oxidative stress, and locomotor dysfunction, have all been displayed by *Drosophila* exposed/treated to neurotoxin or expressing Wt or mutant α-synuclein [54,55]. The *Drosophila* genetic model is an excellent tool to study the function of PD-associated genes and this will be discussed more in detail later in this review.

#### 4.3.3. Zebrafish

Zebrafish have been extensively studied in PD development and pathogenesis [56]. On treatment with neurotoxins, they exhibit altered locomotor activity. These organisms are able to recapture the key biochemical, morphological, neuro–chemical, and behavioral features of PD. Zebrafish gene orthologs to human genes related to PD, have shown to be quite conserved in their function and sequence [1,49].

## 5. PD Induction in Animal Models

Induction of PD in experimental models is achieved by different approaches, including pharmacological intervention, genetic manipulation, or sometimes combination of the two. Here are some insights into these models.

### 5.1. PD Induction in Animal Models by Pharmacological Intervention

The pharmacological models (toxin based) mimic sporadic PD via rapid and increased nigrostriatal dopaminergic loss. Such models can be developed through exposure to neurotoxins, such as 6-OHDA, MPTP, Paraquat, rotenone, etc., or by administration of α-synuclein pre-formed fibril. However, a major limitation observed in such neurotoxin-based models is the absence of the formation of Lewy bodies which is one of the key features of PD [57,58,59,60,61]. Irrespective, animal models of PD induced by the above-mentioned neurotoxin treatments have provided significant knowledge in understanding the disease pathology and identifying potential therapeutic targets.

The different neurotoxin induced animal model systems for PD can be found summarized in Table 1 along with their characteristic features and applications

### 5.2. Commonly Used Neurotoxins to Induce PD

Several compounds are used to induce PD in animal models, and each has advantages and disadvantages. We will discuss them in the following sections:

#### 5.2.1. 6-OHDA (6-Hydroydopamine)

About five decades ago, the first prototype of PD animal modeling was developed by intracerebral administration of 6-OHDA in rats [62]. Since then, it is being used extensively in PD-related research owing to the consistent behavioral phenotype and the degeneration pattern observed in dopaminergic neurons, contributing information regarding biochemical, physiological, and behavioral effects of DA in the central nervous system [62]. While 6-OHDA shows sensitivity in different animals, such as monkeys, mice, dogs, and cats, it is more commonly used in certain species of rats [34,63,64,65]. The addition of an extra-hydroxyl group in DA is the main cause of exhibiting toxicity to dopaminergic neurons. Due to its inability to cross the blood–brain barrier (BBB), this neurotoxin is delivered directly to the target regions, such as the SNpc, the striatum, and the medial forebrain bundle [1,54,56]. Once it reaches the cytoplasm, it rapidly oxidizes to produce ROS-like superoxide radicals, hydroxyl radicals, hydrogen peroxide, etc., all of which eventually lead up to a dysfunctional mitochondrion. Based on the brain region exposed to 6-OHDA, different patterns of neuronal degeneration can be observed [66]. For example, if this neurotoxin is injected into the striatum, it will cause damage to the striatal axon terminals with subsequent dopaminergic neuronal loss in the SNc area [67]. Numerous studies have looked at this type of injection, that causes retrograde neuronal loss, a phenomenon observed in PD patients [68,69,70]. Injection in the SNc proves to be lethal for 60% of TH-containing neurons, followed by loss of the TH-positive terminal in the striatum. Prominent lesion and rapid cellular death are observed upon injection to the SNc as cell bodies of dopaminergic neurons reside in the SNc. Simultaneous to 6-OHDA administration, a norepinephrine transporter (NET) inhibitor (such as desipramine) is given to ensure dopaminergic selectivity (since the toxin can internalize to both dopaminergic and noradrenergic neurons via DAT and NET, respectively) [71].

There are numerous studies investigating the neuro-protective effects of different compounds on 6-OHDA-induced PD animal models [67,72,73,74,75]. While the neurotoxin model does not replicate Lewy body formation or Lewy-like inclusions in the generated PD models, it does exhibit interaction with the α-synuclein protein and has contributed immensely to the field [71]. 6-OHDA finds popularity in its application and use as a potential endogenous toxin in the induction of PD-associated neurodegeneration given that it is a metabolic product of DA (synthesized due to -OH attack and the presence of increased DA levels) [71]. Bilateral injection of 6-OHDA into the striatum results in severe absence of thirst and hunger, which, ultimately, leads to death due to the animal’s inability to care for itself [76]. This caused 6-OHDA to be an excellent component of unilateral modeling of PD [71].

#### 5.2.2. Methyl-4-phenyl-1,2,3,6-tetrahydropyridine (MPTP)

Animal models using MPTP are mainly used to investigate and understand mitochondrial dysfunction in PD. The MPTP animal model is considered a gold standard for a neurotoxin-induced PD animal model due to its ability to replicate almost all of the important hallmarks of PD, such as oxidative stress, ROS, inflammation, and energy failure [1,67,77,78,79]. It is commonly administered to the animal via systemic injection—either subcutaneously or intravenously. In 1982, MPTP was discovered accidentally due to a certain error/issue with its synthesis process. Young drug addicts had intravenously injected these compounds into themselves, only to develop idiopathic parkinsonian syndrome sometime later. Upon investigation into the cause of these symptoms, it was discovered that MPTP resulted in the neurotoxic response with manifestation of parkinsonism [80]. This neurotoxin is a lipophilic molecule allowing it to easily cross the BBB. After systemic administration, MPTP can be oxidized to MPP+ by monoamine oxidase B present in the astrocytes. MPP+ is the main dopaminergic neuro-poisonous compound that confers the observed toxicity [81]. It can be easily taken up/absorbed by the dopaminergic neuron via DAT, due to the structural similarity it shares with dopamine [81]. MPP+ can be transported to synaptic vesicles with the aid of a vesicular monoamine transporter (VMAT) [67], where it induces cell death via inhibition of complex I in the mitochondria. This causes a quick decrease in the ATP concentration in the striatum and the SNpc—resulting in DA neuron apoptosis and necrosis [82]. Additionally, MPP+ displaces the dopamine from the vesicle and enhances the dopamine auto-oxidation, adding more toxicity to neurons. Mice that lack DAT are protected from MPTP toxicity [67]. The most popularly used animal for PD modeling via MPTP is C57/Bl6 mice (I.P. injected) [78,83,84,85]. However, this model is unable to produce Lewy bodies in mice. NHPs on administration with MPTP show similar behavioral and neuro-anatomical properties as that of humans, i.e., they also show bilateral parkinsonian syndrome [67]. The normative practice as seen in some studies initially was treatment of monkeys with high-dose MPTP for a short period of time. These were acute MPTP model systems. However, recent research has identified an MPTP model system that is closer to the human PD pathology via introduction of low-dose MPTP for a longer span of time [86]. Chronic administration of MPTP spanned out over weeks allows the development of a PD model closely resembling the original pathology, except Lewy body formation [87]. Recent applications of the MPTP model system include evaluation of non-motor PD symptoms and conduction of electrophysiological studies that lead to the introduction of deep brain stimulation technology [67].

A certain disadvantage associated with MPTP is the alteration towards a PD-like symptomatic behavior in mice models due to the bilateral lesion. More effort is, thus, required in taking care of the animal’s food and water system. Another drawback is that this toxin, due to some undetermined reason, is insensitive to rats and shows variable sensitivities in different species of mice. Since it only works effectively in C57Bl6 mice, one must be careful when selecting from the subtypes of the mouse model, too [71].

#### 5.2.3. Paraquat (N,N-Dimethyl-4-4-4-bipyridinium)

This commonly used herbicide was identified as neurotoxic agent due to the structural similarity it shares with MPP+. It has been found to exhibit great toxicity to organs, including liver, kidneys, and the lungs [1,88]. A few decades ago, paraquat (PQ) neurotoxicity was tested and investigated in the frog [89]. Findings from the study indicated induction of PD-like behavioral characteristics in the animal. Systemic injection of the pesticide in mice also resulted in dopaminergic neuronal degeneration [57,90,91,92,93]. PQ has the ability to cross the BBB with the assistance of a neutral amino acid carrier (without the requirement of any DAT to enter the dopaminergic neurons). While it bears similarity to MPP+ structure, it does not function by inhibiting mitochondrial complex I. Rather, it alters the redox cycling of glutathione and thioredoxin which impairs the cell’s ability to protect itself from oxidative stress [94]. During the development and characterization of the PD animal model induced by PQ, it was observed that there was a nigral dopaminergic loss without any striatal dopamine depletion, indicating that a different neurochemical pathway maybe guiding the presentation of some of the PD pathology. However, the effect of PQ on nigrostriatal DS system is still under debate as some studies have noted that chronic delivery of PD lead to chronic neurodegeneration and decreased level of DA—a phase that could be adopted to study and understand the pre-clinical stage of PD [95]. Age-dependent nigral dopaminergic neuron loss has been observed and described as a consequence of adult PQ exposure synergistically in combination with other compounds, such as neonatal iron [96]. Other studies, too, have showed how the combination of PQ with other compounds, such as maneb (fungicide)/lactacystin (proteosome inhibitor), results in stronger lesions accompanied by motor deficit and impairments [97,98]. Animals portray decreased motor activity, and dose-dependent loss of striatal TH-positive fibers and neurons in the midbrain SNc [67]. Based on the above-mentioned information, the PQ PD model shows potential in being used to study the earlier disease stages in comparison to other models, given that the PD phenotype appears progressively. Additional investigation and study can help formulate a better understanding of how environmental exposure to such herbicides/pesticides affect PD’s etiology.

#### 5.2.4. Rotenone

Rotenone functions both as a herbicide and an insecticide found naturally in plants [99,100]. Rotenone is a lipophilic compound and can easily cross the BBB where it functions as a mitochondrial complex I inhibitor such as MPTP. It can also inhibit proteasome activity. These result in the development of oxidative and proteolytic stress [88]. However, it leads to a systemic inhibition that is unlike that produced by the use of MPTP [67,101]. There are different routes of rotenone administration that have been tested in animals. One of the first models of Rotenone, described in the year 1985, found that stereotaxic injection of the neurotoxin in high concentrations resulted in significantly lower levels of striatal dopamine and serotonin [102].

This observation is similar to reports on PD where neurodegeneration occurs beyond the dopaminergic system. Rotenone has been found to be related to 35% loss in serotonin, 29% in cholinergic neurons, and 26% reduction in noradrenergic neurons [67]. However, the losses induced by the toxin in such high concentration was not dopaminergic neuron-specific, rather it induced liquefactive necrosis in the striatum. In contrast to this observation, chronic administration of rotenone at lower concentrations induced cell-specific degeneration in the nigrostriatal region. IP injections in animals showed behavioral and neurochemical impairments with a high mortality rate [103,104]. Intravenous (IV) administration has resulted in nigrostriatal DA neuronal damage along with Lewy body-like α-synuclein aggregation, oxidative stress, and gastrointestinal issues [1,105]. PD animal models of rotenone have reported the presence of α-synuclein inclusions in the viable dopaminergic neurons. Other PD-associated characteristics induced by exposure to rotenone include motor impairments, depletion of catecholamine level, and nigral dopaminergic neuronal loss. As of now, continuous administration of low-dose rotenone for a period of about 30 days introduced via novel delivery vehicles has proved to be a good rotenone study model with reduced death rate and robust motor impairments [71]. Given the capability of achieving highly reproducible essential human PD features, researchers have used it to examine and investigate compounds that may have a neuroprotective effect [67,106].

### 5.3. PD Induction in Animal Model by α-Synuclein Pre-Formed Fibril (PFF)

α-synuclein pre-formed fibrils are aggregates of misfolded proteins that are thought to be a major contributor to the development of Parkinson’s disease. These fibrils are formed from the misfolding of α-synuclein proteins, which are found in the brain and other parts of the body. They can cause the death of neurons, which leads to the symptoms of Parkinson’s. These fibrils can also spread between cells, leading to a progressive spread of the disease.

A PFF-induced animal model of α-synuclein is a type of animal model used to study the effects of PFF-like aggregates of the protein α-synuclein on the brain. This type of model is used to study the etiology, pathogenesis, and progression of Parkinson’s disease (PD). In this model, a sample of pre-formed fibrils of α-synuclein is injected directly into the brain of an animal, usually a mouse or a rat. This injection leads to the misfolding of the endogenous synuclein protein and the formation of Lewy bodies, which are the pathological hallmark of PD. Additionally, this model is used to evaluate potential therapeutic strategies for PD. Luk et al. [144] reported successful Parkinson’s-like Lewy pathology by single intra-striatal administration of synthetic α-Syn fibrils. However, this model requires six months to develop neuropathology, such as dopamine neurons loss in the SNc area.

Recently our laboratory developed a PD model combining low-dose MPTP in PFF-injected mouse. It was found that the addition of low-dose MPTP (once daily 10 mg/kg.b.wt for five consecutive days) six weeks after the intra-striatal injection of PFF enhanced the synuclein propagation, increased proteinase K-resistant synuclein filament, and caused significant dopamine neurons death [86]. This model allows the investigator to study the molecular mechanism of synuclein spreading, Lewy body-like pathology, and neuronal death in the SNc area. Additionally, this model can be used to test molecules/inhibitors that can halt synuclein spreading and its associated neuronal demise within a very short period.

### 5.4. PD Induction in Animal Model by Genetic Manipulation

Recently, rarer forms of PD associated with genetic perturbations and mutations in genes encoding for α-synuclein, Parkin, Pink1, and LRRK2 have surfaced and could prove to be potential therapeutic targets. PD genetic models are developed via overexpression of autosomal dominant genes (α-syn and LRKK2) or autosomal recessive genes (knockout or knockdown of genes coding for Parkin, Pink1, and DJ-1) [145]. These have contributed towards establishing and comprehending the molecular mechanisms underlying familial/heritable PD. However, one of the major drawbacks of the genetic model is the failure to produce sufficient dopaminergic neuronal loss, which is the major contributor of PD [1,146].

It is essential to comprehend the underlying mechanisms and principles of the presence of different genetic mutations observed in parkinsonism as they shed light on the shared molecular and biochemical pathways governing both familial and sporadic forms of the disease. The identification of associate pathways in the disease’s progression and pathogenesis contributes greatly to identifying probable therapeutic candidates. While genetic perturbations and mutations observed in PD are quite less, forming only about 10% of diagnosed cases, animal models having a mutated gene are very important because they represent potential therapeutic targets [67]. Linkage and association analyses conducted by different studies for inherited and idiopathic PD, respectively, have identified PD-causing genes. The first to be identified was SNCA (α-synuclein) in several families. Later, other PD-associated genes were discovered that included Parkin, PINK1, LRRK2, and DJ-1 [82,147]. We will discuss some of the mutation models and their importance in the following sections. A summary of the mutation model systems along with their key features can be found in Table 2.

#### 5.4.1. α-Synuclein

This is a small protein of 14 kDa, present in the pre-synaptic terminal regions at a high concentration [1,148]. While the exact function of α-synuclein is yet to be determined, it has been found to regulate activities of the membrane and vesicular organs [149,150]. Mutations, such as substitutions, duplications, and triplications, have been found to be associated in α-synuclein-related genetic disturbances. α-synuclein forms the major component in Lewy bodies observed in PD [112]. Based on this important feature, researchers across communities aim at replicating the PD pathology by overexpression of the WT form of α-synuclein. The first transgenic animal model for α-synuclein was developed in mice by Masliah et al. [151]. Progressive growth and development of neuronal inclusions were seen in the SNc, hippocampus, and neurocortex regions of the brain and they were positively stained for a-synuclein. However, the model did not exactly recapitulate PD pathogenesis as observed in humans, since there was not any significant decrease in dopaminergic neuron, and the α-synuclein inclusions observed did not have a fibrillar structure—therefore it did not resemble Lewy bodies pathology. Subsequently, another α-synuclein model system was developed via TH promoter expression to observe the localized effect of the protein. However, in this case, too, the model failed to achieve α-synuclein inclusion induction and dopaminergic neuronal loss. However, a double mutant (A30P/A53T) model of α-synuclein PD model reported the presence of neurite dystrophy accompanied by motor activity loss and neuronal aggregation formation [71,152].

Mutations caused by the above-mentioned genes result in a dominant inherited PD type [153,154]. A35T mutated mice display a severe motor phenotype that can induce paralysis and animal death [155]. Another phenotype seen to be restricted with mutation in this gene is the resemblance of α-synuclein inclusions to LB [151]. α-synuclein knockout models do not seem to have any effect on dopaminergic neuronal development and maintenance [156,157]. Additionally, α-synuclein knockout mice are resistant to MPTP, suggesting that α-synuclein is a prerequisite for developing PD [157]. Intriguingly, expression of mutant α-synuclein in *Drosophila* models has been observed downregulation of the TH neurons in the SNc, filamentous intraneuronal inclusions, and motor activity impairments [55]. Some α-synuclein transgenic mice models displayed non-motor symptoms similar to deficits in the olfactory system, colonic dysfunction, etc. [158,159]. However, given that the exact role and function of α-synuclein are yet to be understood and identified, its role in PD pathology remains a little puzzling [67].

#### 5.4.2. Leucine-Rich Repeat Kinase 2 (LRRK2)

Mutations in LRRK2 exhibit an autosomal dominant inheritance pattern in familial form of PD [1,160]. They are localized in the membranous region of the cell. The two most commonly observed mutations in this gene comprise G2019S and R1441C/G. The BAC-LRRK2-R1441G transgenic mice models are associated with motor impairments and axonal pathology in the striatal region of the brain—but observed in the absence of dopaminergic neuronal loss and a-synuclein aggregation [161]. Viral vectors, such as herpes simplex virus, or adenoviral vectors have been used to create other types of LRRK2 PD models [162]. Transfection of the mutation LRRK2-G2019S portrayed better stimulation of neurodegeneration and inclusion formation. Infecting HSV-LRRK2-G2019 in the mouse striatum resulted in about 50% reduction of dopaminergic neurons in the SNc [162]. This model offers the opportunity to comprehend complex and essential information regarding the correlation and association between genetic and environmental factors in the pathogenesis and progression of PD [1]. A potential therapeutic option suggested for PD includes the potential of LRRK2 kinase inhibitors—which have been tried and tested [163].

#### 5.4.3. Parkin

Parkin comprises one of the most autosomal recessive patterns of inherited mutation in early-onset PD. It is associated with 50% and 20% of familial and idiopathic types of PD, respectively. Parkin has a ubiquitin ligase and plays an essential part in proteasomal degradation. A loss of function mutation is what results in the disease-causing genetic alteration in its genotype [164]. Parkinsonism stemming from mutation in Parkin could result in aggregation of neurotoxic substrates [1,46]. Knockout (KO) Parkin models have been created to study the effect they have in PD etiology, however, none of the KO models captured the typical PD phenotype [46]. However, Parkin knockout in *Drosophila* causes mitochondrial defects and locomotion deficit, and exhibits reduced life span [165]. On the other hand, Parkin overexpression has been shown to be preventive and protective against dopaminergic neurodegeneration in rats exposed to 6-OHDA or mice treated with MPTP [1,166]. While the application of the gene encoding for this enzyme does not extend very well in PD model construction, it could itself prove to be an essential and potential therapeutic target.

#### 5.4.4. Protein Deglycase (DJ-1)

Mutations affecting the gene encoding for this enzyme are also of the recessive type. Numerous studies have suggested the function of this protein as an antioxidant—that is required on countering the oxidative environment of dopaminergic neurons [167]. However, elimination/downregulation of DJ-1 protein in mice did not induce DA neuronal loss in the SNc, even at an older age. No inclusion bodies were detected, either [168]. What is clear, though, about this protein is that it plays a neuroprotective role in our CNS. DJ-1 KO mice model may be used as an effective means of studying the PD-related molecular mechanism [1,169].

#### 5.4.5. PINK1 (Phosphatase and Tensin Homolog—PTEN-Induced Novel Kinase 1)

PINK1 mutations are also associated with a recessive type of parkinsonism. It is a neuroprotective kinase, mainly found in the mitochondria (intermembrane space) and cytosolic areas. It plays an important role in the differentiation of neurons. Upregulation of PINK1 was found to induce neurite outgrowth in SH-SY5Y neuronal cells along with an increased length of dopaminergic neurons dendrites [170]. Deletion of the PINK1 gene in *Drosophila* causes mitochondrial defects, dopaminergic neuronal degeneration, and locomotor deficits [1,171,172]. Interestingly, in *Drosophila*, knockdown of PINK1 or the Parkin gene displays similar mitochondrial defects and locomotor phenotype [1,165,171,172]. Experiments in PINK1 null flies concluded that Parkin acts downstream of PINK1 [171,172]. However, the PINK1 KO mice model has been shown to be susceptible to oxidative stress and ROS production (without any neuronal degeneration/reduced striatal DA levels, though).

To sum up, none of the above briefly discussed genetic mouse models—while able to efficiently replicate particular aspects of the disease—are able to produce the exact type of neurodegeneration associated with PD and, thus, may need some further/additional modifications/intervention.

### 5.5. PD Induction in Animal Models by Combination of Pharmacological Intervention and Genetic Manipulation

Due to heterogenous manifestation and etiology of PD it is still needed to achieve a single animal model which will demonstrate all the features of PD [3]. However, many researchers are now working on a combination of pharmacological intervention and genetic manipulation [182,183]. This combined approach is, basically, due to two reasons: firstly, some genetic manipulations render the experimental animal more susceptible to neurotoxins for inducing PD [182,184]. Secondly, as genetic factors as well as environmental factors are involved in the etiology of PD, the animals of a combined approach will surely have more PD-like features than the animals of only pharmacological or only genetic intervention [3,49,185].

## 6. Recent Development in PD Model System

An important aspect to be considered in the field of clinical research is that of ‘reproducibility’. This poses a challenge that needs to be tackled with utmost importance and efficiency. One approach can be via improving the quality of experiments conducted in PD research. This can be achieved by constructing and abiding by certain guidelines governing the conduction of pre-clinical PD studies. The other approach comprises of a ‘hypothesis-driven research’. In this, it is important to comprehend the basic biology/physiology of human dopaminergic system function well—at both behavioral and cellular levels [71]. While developing any new model system, it has been observed to achieve decreasing level of α-synucleinopathy and aggregated α-synuclein—as these form an essential part in identifying new PD therapies. α-synuclein levels in blood may provide a reflection of its concentration in the brain and emphasize the requirement of quantifiable targets to monitor the disease progression and clinical outcome [71]. Recently, the advancement in technology assisting the development of stem cell-derived midbrain dopaminergic progenitors resolved issues relating to human neuronal cell availability and ethical issues, allowing PD investigations via the use of DA human neurons in culture [100,186]. Research goals in PD have been shifting towards a more neuroprotective-related focus that offers a better PD study model. DA neurons differentiated from sporadic and inherited PD patient-derived iPS successfully emulate and recapture the PD pathological environment, meaning increased stress, mitochondrial and synaptic abnormalities, pathological accumulation of protein, etc. Another very current development in the PD modeling arena is the potential of midbrain organoids [187,188]. These offer a much better modeling system with pros such as recapturing the glial and neuronal cell interactions [189]. Given that they are a fairly recent development, the drawback associated with them comprises the unavailability of established, robust protocols. While it was demonstrated that human organoids can be translated into the adult mouse brain, they do not provide the opportunity of studying experimental neurorestorative treatments for the impaired motor behavioral phenotype yet. However, midbrain organoids do appear as a promising PD model given the ideal surrounding provided to neuronal cells for growth and development [16]. This strategy was successfully tried and tested in order to achieve development of ‘humanized brains chimeric’ [190].

## 7. Conclusions

To put it in a nutshell, there exist various experimental model systems recapitulating Parkinson’s disease progression and pathogenesis. Its widespread prevalence across the globe and lack of definite treatment or cure necessitates the requirement of either an animal or a cellular model mimicking essential aspects of the disease for researchers to study and understand. These model systems have been developed over several years and are still in the process of improvements and adjustments. With the rising understanding of this disease’s pathology and advancing technology, a combinatorial model looking at both genetic and environmental factors together in human-derived neuronal cells seems to be the future of coming research. In this article, we have summarized some of the existing model systems, their applications, and characteristics. Since PD is thought to be multifactorial, the choice of a model system that should be used to study PD depends on our research objectives and a decision should be reached based on careful study of the strengths and weakness of the experimental model. While a pharmacological model may help determine the effectiveness of drugs to treat PD, a genetic PD model would assist in identifying disease pathway therapeutics targets. Finally, the development of an ideal model system for PD—while it may be a challenging and tedious work in progress—would provide outstanding knowledge, enabling scientists to come up with strong and personalized cures.

## Figures and Tables

**Figure 1 ijms-24-09088-f001:**
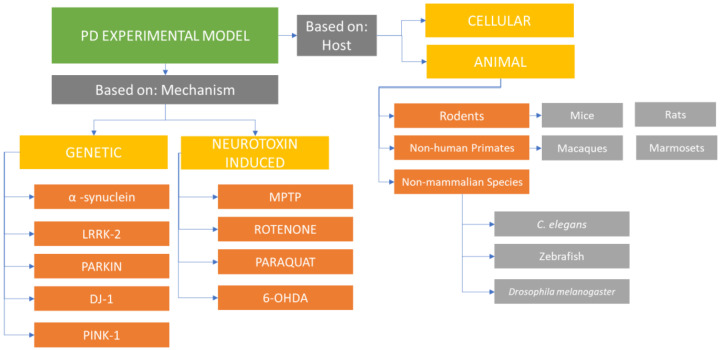
Flowchart summarizing some of the different experimental model types used for Parkinson’s disease.

**Table 1 ijms-24-09088-t001:** Toxin Model System for PD.

Toxin	Mode of Action	Host Species	Key features				Applications	Refs.
			Nigro-Striatal Tract Damage/α-syn Spreading	DA Neuron Loss	Lewy Body-Like Structure	Phenotype/ Motor Symptoms (Behavior)		
6-OHDA	auto-oxidation of 6-OHDA/formation of hydrogen peroxides due to the action of monoamine oxidase/ direct inhibition of mitochondrial respiratory chain complex I	Monkeys (rhesus/ cynomolgus) Mice (Tg/male and female/ C57BL6) Rats (male Wistar/ Fischer 344)	✓ ✓ ✓	✓ ✓ ✓	✓ ✗ ✗	✓ ✓ (deficit in locomotor activity and decrease in motor coordination/ rotational bias) ✓ diminished locomotor activity observed/anxiety- like behavior portrayed.	In general, model systems have been used to investigate neuroprotective effects of different ‘disease modifying strategies’/drugs. They have been used to establish and characterize PD features and develop protocols for the same. Depending upon the objective, factors such as neuroinflammation, various moto/non-motor symptoms are investigated.	[96,97,98,99,100,101,102,103,104,105,106]
MPTP	Mitochondrial complex I inhibition	Mouse (male C57BL6_Tg/Wt) Male Wistar rats Monkey (Macaca fascicularis, Macaca mulatta)	✓ ✓ ✓	✓ ✓ ✓	✓ ✗ ✓	✓ ✓ ✓	Investigate and compare between various MPTP regimens found in the literature. To observe the role of adaptive immune response in PD pathogenesis (the role of the immune system). Observe neuroprotective effects of certain drugs/observe neuroinflammation, cytotoxic effects of microglia and astrocytes. Model system used to investigate chronobiological parameters, and cognitive and motor symptoms upon MPTP administration. Used to observe the relation between MPTP-induced inflammation and gut microbiota, along with any possible differences in PD progression between genders.	[95,107,108,109,110,111,112,113,114,115,116,117,118,119]
ROTENONE	Mitochondrial complex I inhibition	Mice (C57BL6/Swiss) Rats	✓	✓	✗	✓	Observation of the effect of ‘stress’ on disease progression. Studies observed dysfunction in gut –brain access. Use of a lower dose of this neurotoxin to develop a PD model system. Assess effect of social recognition system, GI functioning, and olfactory system. Development of model via environmental contact and investigate underlying pathological and molecular processes. Mostly rotenone rat model system used for looking at neuroprotective effects/ therapeutic interventions.	[59,120,121,122,123,124,125,126,127,128,129,130]
PARAQUAT	Alteration in the redox cycling of ‘glutathione and thioredoxin’	Mice (C57BL6, albino/Tg) Rats (albino male Wistar/Sprague Dawley/male Wistar/long Evans hooded rats	✓	✓	✗	✓	Used to develop a model system to observe neuroprotective effects of pomegranate seed extract and pomegranate juice. One study found intra-nasal administration route showcasing better survivability along with observation of neuronal loss in the SNc and other essential PD-like signs. Generally, the rodents were used to develop a PD model system and identify an underlying molecular mechanism that aids in the disease’s progression.	[131,132,133,134,135,136,137,138,139,140,141,142,143]

✓: the key feature is present; ✗: the key feature is not present.

**Table 2 ijms-24-09088-t002:** Genetic Model Systems for PD.

Gene	Protein	Host System	Key Features				Applications	Refs.
			DA Neuronal Loss	Lewy Body-Like Structure	Phenotype/ Behavior	Mitochondrial Defects		
*SNCA*	α-synuclein	Mice Rat Monkey Zebra fish *C. elegans* *Drosophila*	✓ ✓ ✓ ✓ ✓ ✓	✓ ✗ ✓ ✓ - ✓	✓ ✓ ✓ ✓ ✓ ✓ (climbing defects)	✓	The pathological development of PD takes a long time in this model. However, in the case of *Drosophila* it takes less time for PD development. *Drosophila* model systems are useful for suppressor– enhancer screening.	[1,5,49,56,65,142,146,147,148,149,150,173]
*LRRK2*	Leucine-rich repeat kinase 2	Mice Rat Monkey C. elegans Zebra Fish *Drosophila*	✓ ✗ decreased neuronal viability ✓ - ✓	- - - - - -	✓ ✓ - ✓ Increases in locomotion in adult stages ✓ (climbing defects)	✓ - No mito defect seen, however, increased ROS observed due to increased kinase activity - - -	This model lacks α-synuclein inclusions and dopaminergic neuronal manifestation of PD. This is usually appropriate for LRRK2-specific drug testing.	[1,46,65,162,174,175,176]
*PARKIN*	Parkin	Mice and rat *C. elegans* Zebrafish *Drosophila*	✗ ✓ ✓ ✓	✗	✓ No disturbances in swimming behavior ✓ (climbing defects)	✓ ✓ ✓	*Drosophila* model systems are useful for suppressor–enhancer screening and require less time for PD development.	[1,46,56,65,165]
*PINK1*	PTEN-induced putative kinase protein 1	Mice and rats *C. elegans* Zebra fish *Drosophila*	✓ ✗ disturbed DA projection, no substantial loss of DA neurons. ✓	✗ -	- ✓ ✓ abnormal swimming ✓ (climbing defects)	- ✓ ✓ ✓	*Drosophila* model systems are useful for suppressor–enhancer screening and require less time for PD development.	[1,46,56,65,171,172,177,178]
*DJ-1*	DJ1	Mice Rats *Drosophila* *C. elegans*	✗ ✓ ✓ ✗	✗	✓ demonstrate age-dependent motor deficits of PD. ✓ (climbing defects)	✓	*Drosophila* model systems are useful for suppressor–enhancer screening and require less time for PD development.	[1,46,65,168,177,179,180,181]

## Data Availability

Not Applicable.

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
