# Peer review of "Parkinson’s Disease: Exploring Different Animal Model Systems"

_ijms, 2023, doi:10.3390/ijms24109088_

Round 1

Reviewer 1 Report

The authors summarised the characters of animal models used for Parkinson's disease and commented on the application for different purposes. The work can support a better comprehension of the field and provide good guidance for researchers that want to explore those models in their studies. However, there are a lot of places of comments come directly from other existing review papers. In a review paper, the primary goal is to summarize, synthesize, and critically evaluate the existing literature on a particular topic. While citing other review papers can be helpful in providing a comprehensive understanding of the subject, relying too heavily on them may be considered improper.
The authors should consider reorganizing the following points: 

Line 84, "such as-observed changes" should be "such as observed changes". 

Line 131, the authors should clarify where they cite Figure 1, it's confusing the figure was presented right after line 129.

Line 163,  proper citations should be included for the claim of both macaques and other animals used in the PD model. 

Line 165, α-synuclein alone shouldn't be a causative gene which is a native protein in animals including humans. The authors should make it clear what kind of variants were used. 

Line 154 to 174, the whole NHP discussion only cites 3 previous studies which is too weak for a review.  Citations 1 and 23 were originally reviews and were used for different places talking about specific models. The author should use original research evidence to support the comments. 

Line 210, the same problem of citing an existing review (citation 30) for experiment evidence is not suitable. 

Line 243-246, "The reason why a prominent lesion  and rapid cellular death is observed on injection to nigra while a milder lesion and slower rate of cell death is seen on injection to striatum is the fact that cell bodies of dopaminergic neurons reside in the SNc and their axonal terminal project inside the striatum." The sentence here is not clearly expressed. 

Line 422, "over expression of the wild-type of  mutant form of α-synuclein" is a bit confusing. Could just say "mutant form". 

Line 435-446. The author commented on multiple experimental results yet only one research citation and the rest come from a review paper. 

The language used for this work is proper and easy to understand.  

Author Response

Thank you so much for your fruitful comments and suggestions. Our responses are given below.

The authors summarized the characters of animal models used for Parkinson's disease and commented on the application for different purposes. The work can support a better comprehension of the field and provide good guidance for researchers that want to explore those models in their studies. However, there are a lot of places of comments come directly from other existing review papers. In a review paper, the primary goal is to summarize, synthesize, and critically evaluate the existing literature on a particular topic. While citing other review papers can be helpful in providing a comprehensive understanding of the subject, relying too heavily on them may be considered improper. The authors should consider reorganizing the following points: 

Response: Thank you very much for the critical review of the manuscript and constructive comment. We really appreciate it. We have incorporated the suggestions and have accordingly revised the manuscript.

Q: Line 84, "such as-observed changes" should be "such as observed changes". 

Response: Thanks for the suggestion. We have corrected the sentence accordingly.

Line 131, the authors should clarify where they cite Figure 1, it's confusing the figure was presented right after line 129.

Response: Sorry for the confusion. We have now corrected the position of Figure 1 and added the proper citation.

Line 163, proper citations should be included for the claim of both macaques and other animals used in the PD model. 

Response: Thanks for the suggestion. We have added the citations related to macaques and other animals used in the PD model. (citations 35-39)

Line 165, α-synuclein alone shouldn't be a causative gene which is a native protein in animals including humans. The authors should make it clear what kind of variants were used. 

Response: Thanks for the excellent comment. We have explained it more clearly in the respective section.

Line 154 to 174, the whole NHP discussion only cites 3 previous studies which is too weak for a review.  Citations 1 and 23 were originally reviews and were used for different places talking about specific models. The author should use original research evidence to support the comments. 

Response:  We appreciate the reviewer’s comment. We have added the citation from the original work. (citations 35 to 39 and 40 to 46)

Line 210, the same problem of citing an existing review (citation 30) for experiment evidence is not suitable. 

Response: Thank-you again. We have replaced the citation with original work. (citations 57-61)

Line 243-246, "The reason why a prominent lesion and rapid cellular death is observed on injection to nigra while a milder lesion and slower rate of cell death is seen on injection to striatum is the fact that cell bodies of dopaminergic neurons reside in the SNc and their axonal terminal project inside the striatum." The sentence here is not clearly expressed. 

Response: Thank you for the comment. We have corrected the sentence as “Prominent lesion and rapid cellular death is observed on injection to the SNc as cell bodies of dopaminergic neurons reside in the SNc” in the revised manuscript.

Line 422, "over expression of the wild-type of mutant form of α-synuclein" is a bit confusing. Could just say "mutant form". 

Response: Thank you for highlighting. This sentence is corrected to “over expression of the wild-type form of α-synuclein" in the revised manuscript.

Line 435-446. The author commented on multiple experimental results yet only one research citation and the rest come from a review paper. 

Response:  Thanks for the suggestion. We have added the citations. (citations 55, 126, 128, 129-134)

Reviewer 2 Report

A study in a clear way outlining the selected, interesting research topic.

Author Response

A study in a clear way outlining the selected, interesting research topic.

Response: Thank you very much for your appreciation!

Reviewer 3 Report

This review manuscript describes how different animals models have been used to study PD. The manuscript is well written but can be improved. Below are some of my comments for the manuscript.

1) The section of animal models needs to be considerably improved. The mouse model only describes the behavioral outcomes but does elaborate on the similarities at a molecular level. The entire section seems a bit weak and needs more description.

Author Response

Thank you so much for your fruitful comments and suggestions. Our responses are given below:

This review manuscript describes how different animals models have been used to study PD. The manuscript is well written but can be improved. Below are some of my comments for the manuscript.

The section of animal models needs to be considerably improved. The mouse model only describes the behavioral outcomes but does elaborate on the similarities at a molecular level. The entire section seems a bit weak and needs more description.

Response: Thank you very much for the excellent suggestion. We have edited the section extensively. Please see the changes as indicated by yellow highlight.

Round 2

Reviewer 1 Report

The authors revised the manuscript appropriately, which can be accepted for publication.